# Automatic Measurement of Seed Geometric Parameters Using a Handheld Scanner

**DOI:** 10.3390/s24186117

**Published:** 2024-09-22

**Authors:** Xia Huang, Fengbo Zhu, Xiqi Wang, Bo Zhang

**Affiliations:** 1School of Electronic Engineering, Chengdu Technological University, Chengdu 611730, China; yuxiuhx@163.com; 2Special Robot Application Technology Research Institute, Chengdu 611730, China; 3School of Geospatial Information, Information Engineering University, Zhengzhou 450001, China; zhu_fb@whu.edu.cn; 4School of Transportation Engineering, Shandong Jianzhu University, Jinan 250101, China; wangxiqi24@sdjzu.edu.cn

**Keywords:** seed, parameter measurement, handheld scanners, point cloud, 3D models

## Abstract

Seed geometric parameters are important in yielding trait scorers, quantitative trait loci, and species recognition and classification. A novel method for automatic measurement of three-dimensional seed phenotypes is proposed. First, a handheld three-dimensional (3D) laser scanner is employed to obtain the seed point cloud data in batches. Second, a novel point cloud-based phenotyping method is proposed to obtain a single-seed 3D model and extract 33 phenotypes. It is connected by an automatic pipeline, including single-seed segmentation, pose normalization, point cloud completion by an ellipse fitting method, Poisson surface reconstruction, and automatic trait estimation. Finally, two statistical models (one using 11 size-related phenotypes and the other using 22 shape-related phenotypes) based on the principal component analysis method are built. A total of 3400 samples of eight kinds of seeds with different geometrical shapes are tested. Experiments show: (1) a single-seed 3D model can be automatically obtained with 0.017 mm point cloud completion error; (2) 33 phenotypes can be automatically extracted with high correlation compared with manual measurements (correlation coefficient (*R*^2^) above 0.9981 for size-related phenotypes and *R*^2^ above 0.8421 for shape-related phenotypes); and (3) two statistical models are successfully built to achieve seed shape description and quantification.

## 1. Introduction

Seeds are a fundamental form of survival, propagation, and reproduction for plants [1,2,3]. Seed phenotypes, including size- and shape-related phenotypes, are important in species recognition and classification [4], quality evaluation [5], optimization breeding [6], and yield evaluation [7]. The volume, surface area, length, width, thickness, cross-sectional perimeter, and area are size-related phenotypes [8,9]. The roundness, needle degree, flatness, shape factor, sphericity, elongation, circularity, and compactness are shape-related phenotypes [10,11,12,13]. Size-related phenotypes are often used for seed conditions and quality detection. Shape-related phenotypes are dimensionless and insensitive to the seed size, which is important in seed species recognition and classification. The traditional manual measurement of seed geometric parameters is no longer suitable for the needs of smart agriculture [14,15]. Therefore, it is meaningful to explore a novel method of seed geometric parameters automatic measurement.

High-throughput phenotyping is changing the conventional agricultural measurement methods [16]. It promotes the quantitative trait loci (QTL) study [17] and taxonomic analysis [18]. The phenotypes can be measured using different methods for different applications. The conventional manual measurement method using a vernier caliper can obtain the seeds’ length, width, and thickness. The manual measurement method is time-consuming, costly, and relies on manual experience, which is suitable for research involving a small number of samples [19]. Machine vision technology is often used for automatic measurement. The length, width, and projected perimeter and area can be obtained [9,13], and the elongation, circularity, and compactness can also be extracted using 2D images. It should be noted that these phenotypes are usually calculated based on the seed’s main projection profile (the maximum principal component profile) while the other two profiles, which are also important, are almost not discussed. Since the thickness, 3D volume, and 3D surface area cannot be acquired with 2D digital images, 3D technology has become an active area of research for seed automatic measurement. Structure from motion (SFM) is a classic way to obtain a 3D point cloud [20]. However, the data acquisition using the SFM method is usually consuming. 3D laser scanning outperforms in data acquisition. Li et al. [21] collected the single seed from four viewpoints using a 3D laser scanning system and registered the obtained point cloud using Geomagic Studio software to obtain the complete 3D point cloud of a single rice seed. Length, width, thickness, and volume were automatically extracted. However, the data acquisition of the seed point cloud was time-consuming, which was unsuitable for batch data acquisition for a large number of samples. Yan [22] adopted a Konica Minolta Vivid 910 3D scanner to obtain the single corn point cloud. The data were obtained in batches while the 3D reconstruction was completed by the software Geomagic Studio. The automation should be improved. Liu et al. [8] used X-ray computed tomography (CT) scanning to automatically obtain seed point clouds in batches and extract the length, width, thickness, radius, volume, surface area, compactness, and sphericity. However, they ignored the problem that the obtained 3D models were incomplete because the bottom part of the seed facing the table was not scanned. 

The current active areas of research for seed automatic phenotyping using 3D scanners are batch data acquisition, 3D reconstruction of a single seed, and phenotype estimation. The difficulty is to achieve automatic measurement and batch data processing simultaneously [23]. One of the key problems is to obtain the complete 3D model of a single seed based on the obtained incomplete point cloud by a novel method, namely, to explore an effective and robust approach for point cloud completion. Kazhdan et al. [24] adopted an implicit fitting strategy to obtain a complete shape surface. It is only suitable for cases where a small number of point clouds are missing and easy to overfit. Zhang et al. [25] used an object retrieval method to obtain the complete point cloud. It relies on the huge number of models in the database and the rich model types. It is difficult to complete models that are not in the database. Zhang et al. [26] introduced generative adversarial network (GAN) inversion to shape completion. It depends on the input database and has a limitation on the storage space of the graphics processing unit (GPU), which will affect both the measurement accuracy and the four processing times. Thus, it is necessary to study an effective and practical point cloud completion method. 

Size- and shape-related phenotypes can be calculated based on the single-seed 3D model. Seed shape description and quantification will be more adequate with more numbers and types of phenotypes. Recent methods for seed shape description and quantification use geometric models to represent seed shapes [23,27], which combine computer vision technologies with statistical algorithms. They may be efficient and robust for seed discrimination in a range of plant species and varieties. However, various geometric models are required for different types of seeds with different shapes. Typical geometric models are sphere, ellipse, oval, heart-shaped, kidney-shaped, cardioid, Fibonacci’s spiral contour, and lenses of varied proportions [28]. However, it is difficult for geometric models to describe irregular geometry seeds such as broad beans and peanuts. Thus, it is important to build a unified shape model to describe different seeds with different shapes. The statistical model is a useful tool to quantitatively describe an object’s shape [29]. Various 2D phenotypes have been used for building statistical shape models [30,31,32], while 3D phenotypes were rarely used, and the number and types of phenotypes were limited. Therefore, it is meaningful and practical to explore a unified seed shape description and quantification method using statistical shape models based on phenotypes derived from a single-seed 3D model.

In our previous work, a symmetry-based 3D reconstruction method is proposed, which is only suitable for seeds with symmetrical shapes [33]. A more effective and robust point cloud completion method will be handled in this work. The purpose of this work is to achieve seed automatic phenotyping. Key problems in point cloud-based phenotyping research, namely, batch four acquisition and processing, point cloud completion, and automatic phenotype estimation, will be handled. The batch data acquisition for seeds’ point clouds is designed to be conducted by a handheld laser scanner (RigelScan Elite). An automatic cloud-based phenotyping pipeline of single-seed segmentation poses normalization, point cloud completion by a least-squares ellipse fitting method, Poisson surface reconstruction, and automatic trait estimation is proposed. Two statistical models (one using 11 size-related phenotypes and the other using 22 shape-related phenotypes) based on the principal component analysis method are built. Eight kinds of seeds with 3400 samples will be tested. The experiment objects are broad beans, peanuts, pinto beans, soybeans, black beans, red beans, and mung beans. The main contribution of this paper is to propose an automatic phenotyping method and achieve shape description and quantification by size- and shape-related statistical models.

## 2. Materials and Methods

The flowchart for automatic measurement of seed geometric parameters based on handheld scanners is shown in Figure 1. First, a handheld 3D laser scanner is used for batch 3D point cloud acquisition. Second, an automatic point cloud-based phenotyping pipeline, including point cloud processing and phenotype estimation, is proposed. Point cloud processing mainly includes single-seed segmentation, poses normalization, point cloud completion, and surface reconstruction. Specifically, the single-seed segmentation is realized by combining random sample consensus (RANSAC) segmentation method, the region growing segmentation method, and the dimension feature detection method. Poses normalization adopts the principal component analysis (PCA) rotation method. Point cloud completion is implemented using an ellipse fitting-based point cloud completion method. Surface reconstruction is conducted using the Poisson surface reconstruction method. A single-seed 3D model and 33 phenotypes, including 11 size-related phenotypes and 22 shape-related phenotypes, can be automatically obtained. Finally, two statistical models, one using size-related phenotypes and the other using shape-related phenotypes are established based on the PCA method to implement the shape description and quantification. The weights of size- and shape-related phenotypes are discussed. 

### 2.1. Data Acquisition

The data acquisition obtained the seeds’ 3D point cloud data. Dry broad bean, peanut, pinto bean, soybean, black bean, red bean, and mung bean seeds were used as experiment objects. The materials were bought from the market and were of good quality and each kind had uniform samples of similar size and shape. No shriveled seeds were involved. Data acquisition was performed using a handheld 3D laser scanner (RigelScan Elite, made by Zhongguan Automation Technology Co., Ltd., Wuhan, China) indoors, in Wuhan, China, in January 2022. The RigelScan Elite scanner’s working principle is the triangulation method. RigelScan Elite scanner has a laser light-emitting diode and two visible-light cameras. The laser diode can emit three modes of laser, namely 11 pairs of laser line modes for rough scanning, 5 pairs of laser line modes for small area scanning, and 1 deep-hole mode for detailed scanning. The laser lights are focused and projected onto the target through a lens and the reflected or diffused laser lights are imaged by the two visible-light cameras. The location and spatial information of the target are obtained according to a certain triangle relationship. 

The seeds of broad beans, peanuts, pinto beans, soybeans, black beans, red beans, and mung beans were on the table and scanned in batches. A total of 300 samples for broad beans, peanuts, and pinto beans and 500 samples for soybeans, black beans, red beans, and mung beans were used because of the limitation of the table area. A few samples were sticking together and there were no heavily overlapping and pasting seeds. Each kind was scanned once. The seeds were on the table and scanned using RigelScan Elite (Figure 2a). The scanning angles were arbitrary and random. Some reflective markers were pasted on the table before scanning, which was used for point cloud stitching between multiple frames (Figure 2c). The frame rate was 1,050,000 times/s and the field depth was 550 mm. The scanning process was monitored from the computer in real time as it was performed. 

The obtained point cloud can be directly output from the computer after the data scanning, as shown in Figure 2e (filtering 50% for effective visualization because of 0.01 mm cloud density). As shown in Figure 2d, the output point cloud has no color information and the point cloud of the seed bottom part is incomplete because one side of the seeds is facing the table. The scanned point cloud includes the table and seeds data. 

### 2.2. Point Cloud-Based Phenotyping

#### 2.2.1. Point Cloud Processing

The point cloud processing should be handled before the extraction of phenotypes, which is designed to obtain a single-seed 3D model. It mainly includes single-seed segmentation, pose normalization, point cloud completion, and surface reconstruction. 

(1) Single-seed segmentation

Single-seed segmentation is to extract a single-seed point cloud from the background (table point cloud). The RANSAC [34] plane detection method is adopted to remove the table points. The distance threshold here is 0.05 mm. Therefore, most table points are removed while some table side points are reserved, as shown in Figure 3b. Then the region-growing segmentation method [35] based on smooth and curvature characters is used to extract the single-seed point clouds. Here the smoothness threshold is 150°/180π and the curvature threshold is 0.4 by experience. A series of clusters is obtained. Some clusters of the table edge points are reserved, as shown in Figure 3c (details seen in Figure 3e). Noticing that the reserved table points clusters are planar and the single-seed point clouds are three-dimensional. Then the dimensional features [36] are used to remove these preserved table points. Performing principal component analysis processing on each cluster, the eigenvalues of the three principal component dimensions *λ*_1_, *λ*_2_, and *λ*_3_ (*λ*_1_ ≥ *λ*_2_ ≥ *λ*_3_) can be obtained. The dimension features of each point cloud are calculated as follows:(1)$$a_{1D}=\frac{\sqrt{\lambda_{1}-\lambda_{2}}}{\sqrt{\lambda_{1}}},\;a_{2D}=\frac{\sqrt{\lambda_{2}-\lambda_{3}}}{\sqrt{\lambda_{1}}},\;\mathrm{and}\;a_{3D}=\frac{\sqrt{\lambda_{3}}}{\sqrt{\lambda_{1}}},$$
where *a*_1D_, *a*_2D_, and *a*_3D_ are a one-dimensional linear feature, two-dimensional planar feature, and three-dimensional scattered point feature, respectively. The sum of *a*_1D_, *a*_2D_, and *a*_3D_ is 1. The point clouds are classified as linear, planar, or three-dimensional based on the relationship of *λ*_1_, *λ*_2_, *λ*_3_, *a*_1D_, *a*_2D_, and *a*_3D_. A point cloud is linear when *a*_1D_ is larger than *a*_2D_, and *a*_3D_ and λ_1_ >> λ_2_ ≈ λ_3_ simultaneously. It is planar when *a*_2D_ is larger than *a*_1D_, and *a*_3D_ and λ_1_ ≈ λ_2_ >> λ_3_. It is three-dimensional when *a*_3D_ is the largest among *a*_1D_, *a*_2D_, and *a*_3D_ and λ_1_ ≈ λ_2_ ≈ λ_3_. The point clouds with the planar feature are removed and those with a three-dimensional scattered point feature are reserved. These reserved point clouds are seed point clouds, as shown in Figure 3d (details seen in Figure 3f). 

(2) Pose normalization

It is necessary to normalize the measurement pose of individual seeds in the world coordinate system to simplify the calculation of the seeds’ kernel traits. Here, the coordinate rotation is conducted using the PCA method:(2)$$\left[\begin{array}{c} x\\ y\\ z \end{array}\right]=\left[e_{v1},e_{v2},e_{v3}\right]\left[\begin{array}{c} x_{0}\\ y_{0}\\ z_{0} \end{array}\right]$$
where **e**_v1_, **e**_v2_, and **e**_v3_ are three eigenvectors in descending order by performing PCA processing on seed point clouds, respectively. The measurement poses of individual seeds in the world coordinate system are the same after the coordinate rotation: (1) the geometric centre of the scanned seed point cloud overlaps the origin of the world coordinate system; (2) and the seed’s length, width and thickness directions are the *X*-, *Z*- and *Y*-axis directions of the world coordinate system, respectively, as shown in Figure 3g,h. Therefore, the first, second, and third principal component profiles are the three main important profiles for seed shape description and quantification in plant research.

(3) Point cloud completion

It can be observed that the obtained single-seed point cloud is incomplete and lacks the data of the seed facing the table, as shown in Figure 2d. The challenge is to obtain a complete 3D model from the incomplete scanned point cloud, namely, to explore an effective and robust point cloud completion method. Seeds are rigid and their longitudinal profiles are approximately ellipsoidal [18,37]. Figure 4 shows the seed longitudinal profile (YOZ) contour fitted by the B-spline curve, circle, and least-squares ellipse. For seeds with irregular geometric shapes, such as broad beans, peanuts, and kidney beans, the least-squares ellipse fitting based on the longitudinal profile points is constructive and robust, while the B-spline curve and circle fitting fail. For seeds with regular geometric shapes, such as soybeans, peas, black beans, red beans, and mung beans, all three fitting methods are applicable. The ellipse fitting results outperform those of the B-spline curve and the circle fitting. Therefore, this paper exploits the seed ellipsoidal profile characteristic to conduct 3D point cloud completion.

The 3D point cloud completion method proposed in this paper mainly includes four steps. Let the single-seed point cloud be denoted *PC*. First, a series of sliced point clouds, namely the scanned incomplete profile point clouds, *PC*_1_, *PC*_2_, …, *PC*_20_, is obtained by dividing *PC* into 20 pieces along the *X*-axis, as shown in Figure 5a. Here, *PC* = {*PC*_1_, *PC*_2_, …, *PC*_20_}. Second, an ellipse is fitted using a sliced point cloud (*PC_i_*) (Figure 5b) in *PC* using the least square fitting ellipse method [38] based on the incomplete profile point cloud *PC_i_*, as shown in Figure 5c. This ellipse fitting method incorporates the ellipticity constraint into the normalization factor by minimizing the algebraic distance subject to the constraint 4*ac* − *b*^2^ = 1. Since this ellipse fitting method is ellipse-specific, even heavily scattered points will always return an ellipse. In addition, it can be solved effectively and practically by a generalized eigensystem. Third, a complete profile point cloud *PR_i_* is filled by *PC_i_* (red points in Figure 5d) and parts points of the fitted ellipse *PE_i_* (blue points in Figure 5d). Here *PR_i_* = {*PC_i_*, *PE_i_*}. Then, a series of profile point clouds, *PR*_1_, *PR*_2_, …, *PR*_20_, is automatically obtained. The complete point cloud of single-seed *PR* is reconstructed and *PR* = {*PR*_1_, *PR*_2_, …, *PR*_20_}, as shown in Figure 5e–g. Finally, the reconstructed point cloud is filtered by the Gaussian filtering method [39] for uniform point cloud density, as shown in Figure 5h. Gaussian filter is a nonlinear filtering method that uses the principle of weighted average to filter point clouds, which can effectively smooth and denoise point clouds.

(4) Surface reconstruction

The classic Poisson surface reconstruction method [40], which has the advantages of global and local fitting, is adopted to measure seeds’ volume and surface area. It is based on the Poisson equation, which is derived from the Laplace equation, namely the potential equation calculated by
(3)$$\frac{\partial^{2}\varphi }{\partial x^{2}}+\frac{\partial^{2}\varphi }{\partial y^{2}}+\frac{\partial^{2}\varphi }{\partial z^{2}}=0,$$
where *x*, *y*, and *z* are the coordinate values of the points, and *φ* is a real-valued function that is twice differentiable in *x*, *y*, and *z*. The Poisson equation introduces a concept related to gravity, that is, if there is an observation point *P* in a space filled with gravitational medium, the Laplace equation can be changed to
(4)$$\frac{\partial^{2}\varphi }{\partial x^{2}}+\frac{\partial^{2}\varphi }{\partial y^{2}}+\frac{\partial^{2}\varphi }{\partial z^{2}}=-4\pi G\rho ,$$
where *ρ* is the mass density and *G* is the gravity acceleration. The Laplace equation can be transformed as follows through mathematical identity transformation:(5)Δφ=−4πGρ,
where Δ represents the Laplace operator and the Poisson equation is modified:(6)Δφ=f.
where *f* is the mass distribution of the gravitational field.

Figure 5i,j show the triangle mesh, and surface visualization of one peanut’s 3D model fitted by the Poisson surface reconstruction method.

#### 2.2.2. Phenotype Estimation

Seed volume (*V*) and surface area (*S*) are important 3D size-related phenotypes in shape description and quantification [8,10,11,17]. *V* can be regarded as the volume of a closed space enclosed by a triangular mesh, namely the sum of the projected volumes of all signed triangular patches. *V* can be regarded as the total surface area of the triangular mesh and *S* can be regarded as the total surface area of the triangular mesh (Figure 6a). Length (*L*), width (*W*), and thickness (*T*) are three directly size-related phenotypes for size description and quantification [3,9,10,11,29]. Here *L*, *W*, and *T* are computed using the AABB box algorithm. Specifically, *L*, *W*, and *T* can be seen as the length, width, and height of the AABB box (Figure 6a). Perimeter (*C*) and area (*A*) of the cross-section are important size-related phenotypes for seed profile shape description and quantification [3,8,9,10]. Three principal component profiles that are detected by performing *PCA* processing on the reconstructed point cloud are the most important three main profiles in seed shape analysis. These three principal component profiles are perpendicular to each other across the seed center (Figure 6b). specifically, *C*_1_, *C*_2_, *C*_3_, *A*_1_, *A*_2_, and *A*_3_ are the perimeter and area of the first, second, and third profiles. The details of calculation algorithms for *V*, *S*, *L*, *W*, *T*, *C*_1_, *C*_2_, *C*_3_, *A*_1_, *A*_2_, and *A*_3_ can be seen in our previous work [33]. 

According to the related research [9,11,41,42], 11 size-related phenotypes and 22 shape-related phenotypes are automatically calculated in this paper, as listed in Table 1.

Shape-related phenotypes are dimensionless and insensitive to region size. The radius ratio (*RR*) is an important feature that identifies the seed shape [10]. Here, *RR* = *d*_MAX_/*d*_MIN_, where d_MAX_ and d_MIN_ are points of seeds with the maximum and minimum distance from the center of mass. Roundness (*R*) describes the sharpness of the corners [11]. The needle degree (*ND*), flatness (*F*), shape factor (*SF*), and sphericity (*SP*) of seeds are the quantitative indexes of the seeds [10]. Elongation (*E*), circularity (*CR*), and compactness (*CP*) are used to describe profile shape features. Elongation is the difference between the lengths of the major and minor axes of the best-fit ellipse divided by the sum of lengths. It is zero for a circle and one for a long and narrow ellipse. Circularity indicates the compactness of a region to a certain degree. Compactness describes the resemblance of the object to a square shape. The value of compactness will be one for a perfect square. The bounding rectangle to the perimeter (*BR*) indicates the convex feature of the profile. Its value will equal one for a convex shape object. The geometric mean (*D*) [11] is calculated by considering the spherical shape of a seed. Here, *D* = (*LWT*)^1/3^.

### 2.3. Shape Description and Quantification Based on Statistical Models

The statistical shape model based on the PCA method is a useful tool to quantitatively describe an object’s size and shape [43]. The principal component is to recombine multiple variables (*X*_1_, *X*_2_, …, *X_i_*) with strong linear correlation to generate a few variables (*F*_1_, *F*_2_, …, *F_i_*) that are not related to each other so that they can extract as much information as possible from the original variables. *F*_1_, *F*_2_, …, *F_i_*, are called principal components, which are the first, second, …, and the *i*-th principal component, respectively. The process of building a statistical shape model based on the PCA method is as follows:

(1) Construct a sample matrix
(7)$$X=\left[\begin{array}{cccc} x_{11} & x_{12} & \ldots & x_{1p}\\ x_{21} & x_{22} & \ldots & x_{2p}\\ \vdots & \vdots & \vdots & \vdots \\ x_{n1} & x_{n2} & \cdots & x_{np} \end{array}\right],$$
where *x_ij_* represents the value of the *j*-th variable in the *i*-th group of sample data.

(2) Transform the sample matrix
(8)$$Y={\left[y_{ij}\right]}_{n\times p},$$
where $y_{ij}=\{\begin{array}{cc} x_{ij}, & \mathrm{for}\;\mathrm{positive}\;\mathrm{index}\\ -x_{ij}, & \mathrm{for}\;\mathrm{negative}\;\mathrm{index} \end{array}$

(3) Perform a standardized transformation on *Y* so that *Y* becomes a standardized matrix **Z**
(9)$$Z=\left[\begin{array}{cccc} z_{11} & z_{12} & \ldots & z_{1p}\\ z_{21} & z_{22} & \ldots & z_{2p}\\ \vdots & \vdots & \vdots & \vdots \\ z_{n1} & z_{n2} & \cdots & z_{np} \end{array}\right],$$
where z_ij_ = (*y_ij_ − y_j_*)/*s_j_*, *y_j_*, and *s_j_* are the mean and standard deviation of the *j*-th column in the *Y* matrix, respectively.

(4) Calculate the sample correlation coefficient matrix of the standardized matrix **Z**
(10)$$R=\frac{Z^{T}Z}{n-1},$$

(5) Find the eigenvalues
(11)$$\left|R-\lambda I_{p}\right|=0,$$
where **I***_p_* is the identity matrix of row *p* and column *p*, λ is characteristic value and λ_1_ > λ_2_ > … > λ*_p_* > 0.

(6) Calculate the value of *m* to make sure the utilization rate of information reaches more than 80%.
(12)$$\frac{\sum_{j=1}^{m}\lambda_{j}}{\sum_{j=1}^{p}\lambda_{j}}\geq 0.8(j=1,2,\ldots ,p),$$

(7) Find the principal components of z*_i_* = (z*_i_*_1_, z*_i_*_2_, …, z*_ip_*)^T^
(13)$$u_{ij}=z_{i}^{T}b_{j}^{0}(j=1,2,\ldots ,m),$$
where $b_{j}^{0}$ is the variance explanatory power of the *j*-th principal component. Then calculate the decision matrix **U**
(14)$$U=\left[\begin{array}{cccc} u_{11} & u_{12} & \ldots & u_{1m}\\ u_{21} & u_{22} & \ldots & u_{2m}\\ \vdots & \vdots & \vdots & \vdots \\ u_{p1} & u_{p2} & \cdots & u_{pm} \end{array}\right],$$
where *u_ij_* is the *j*-th principal component vector of the *i*-th variable.

(8) Assuming that the number of indicators with undetermined weights is *w_i_*, establish a primary weight model, namely the principal component model
(15)$$\{\begin{array}{c} F_{1}=u_{11}w_{1}+u_{21}w_{2}+\cdots +u_{L1}w_{L}\\ F_{2}=u_{12}w_{1}+u_{22}w_{2}+\cdots +u_{L2}w_{L}\\ \vdots \\ F_{m}=u_{1m}w_{1}+u_{2m}w_{2}+\cdots +u_{Lm}w_{L} \end{array},$$
where **F**_1_, **F**_2_, …, and **F***_m_* are the *m* principal components obtained after analysis; *u_ij_* are the coefficients in the decision matrix. Establish a comprehensive evaluation function
(16)$$F_{z}=a_{1}v_{1}+a_{2}v_{2}+\cdots +a_{L}v_{L},$$
where *a*_1_, *a*_2_, …, *a_L_* are the comprehensive importance of indicators and *v*_1_, *v*_2_, …, *v_L_* are the features. *F_z_* is the statistical shape model based on the PCA method. Here two statistical shape models, one for seed size based on morphological traits and the other for seed shape based on shape features, are built for quantitative description of seeds. It should be noted that the sum of *a*_1_, *a*_2_, …, *a_L_* is beyond 1. To normalize the weights, the comprehensive value of the original index score can be calculated
(17)$$V_{zi}=\sum_{j=1}^{L}a_{j}p_{ij},$$
where *p*_ij_ is the comprehensive score model coefficient and *i* = 1, 2, …, *h*. Therefore, the weight of each indicator is
(18)$$w_{i}=\frac{V_{Zi}}{\sum_{i=1}^{h}V_{Zi}}.$$

### 2.4. Accuracy Analysis

Data scanning, segmentation, 3D point cloud completion, surface reconstruction, and trait estimation are the main factors affecting measurement accuracy. The scanning accuracy *R*__*scan*_ is usually measured by the ratio of scanned seeds number *N*_2_ and total seeds number *N*_1_: *R*__*scan*_ = *N*_2_/*N*_1_ × 100%. The segmentation accuracy *R*__*seg*_ is usually measured by the ratio of extracted seeds number *N*_3_ and the input seeds number, namely, scanned seeds number *N*_2_: *R*__*seg*_ = *N*_3_/*N*_2_ × 100%. 

The reconstruction error, defined as the average distance between the true point and the closest reconstructed point, is used to verify the accuracy of the 3D point cloud completion. The detail of the reconstruction error can be seen in the paper [33]. 

There will be a certain error between the completed point cloud and the real point cloud of the seed. This paper compares the completed point cloud with the real seed point cloud to calculate the point cloud completion error. The real point cloud of the seeds is selected from 20 seeds of each kind, fixed on the desktop with a needle, and scanned 360 degrees with RigelScan Elite. Each seed is scanned separately to obtain a complete real seed point cloud. *E_r_* is used to represent the completion error:(19)$$E_{r}=\frac{1}{n_{p}}\sum_{i=1}^{n}d(P_{ci},P_{gj}),$$
where *n_p_* is the real point cloud number and *d*(*P_ci_*, *P_gj_*) is the distance between point *P_ci_* (a point in the completed point cloud) and its closest point *P_gj_* (a point in the real point cloud).

Regression analysis between the automatically measured phenotypes and the manually measured values is conducted to illustrate the trait measurement accuracy. A total of 20 samples of each kind (the same seeds for 3D reconstruction accuracy verification) were measured manually. The ground truths of length, width, and thickness were obtained using a vernier caliper. The other traits were measured by the software Geomagic Studio based on the real 3D point cloud. All the ground truths were manually measured three times by three people, and the average was adopted.

## 3. Results

### 3.1. Data Scanning and Segmentation Results

The seed point clouds can be obtained in batches using a handheld 3D laser scanner (RigelScan Elite). The single-seed point cloud can be automatically extracted with our proposed segmentation method. Figure A1 shows the scanning and segmentation results of broad beans, peanuts, pinto beans, soybeans, black beans, red beans, and mung beans. The obtained point clouds are rendered for effective visualization (Figure A1b) while they are with no color information. They are incomplete with no data facing the table part (Figure A1d). It illustrates that the seeds are well scanned, and the single seed, including the sticking seeds (some samples in Figure A1e), can be successfully extracted. 

Table 2 lists the numbers of the total successfully scanned and segmented seeds and the corresponding scanning accuracy and segmentation accuracy. It shows that 3396 samples from 3400 samples were successfully scanned. The scanning accuracies of broad beans, peanuts, soybeans, black beans, and mung beans are 100%, while the scanning accuracies of pinto beans and red beans are 99.67% and 99.40%, respectively. The average scanning accuracy of all the kinds of seed is 99.88%, and the corresponding average segmentation accuracy is 100%. The high scanning accuracy proves the handheld 3D laser scanner (RigelScan Elite) is effective for point cloud data acquisition in batches. The high segmentation accuracy verifies that the proposed segmentation method is practical and robust for obtaining single-seed point clouds.

### 3.2. Point Cloud Completion Results

The complete 3D model of a single seed based on the incomplete scanned point clouds can be directly obtained using an ellipse-based point cloud completion method. Figure 7 shows 3D point cloud completion results. It shows that the obtained seed point clouds are incomplete. Our reconstructed point clouds are complete and well reconstructed and the corresponding 3D mesh models are well fitted. The sizes of broad beans, peanuts, pinto beans, soybeans, black beans, red beans, and mung beans are in descending order. Broad beans, peanuts, and pinto beans have irregular geometrical shapes, whereas soybeans, peas, and black beans have spherical seeds, and red and mung beans have ellipsoid seeds.

Figure 8 presents the 3D point cloud completion errors. The mean values of 3D reconstruction errors for broad beans, peanuts, pinto beans, soybeans, black beans, red beans, and mung beans are 0.023, 0.021, 0.018, 0.014, 0.016, 0.016, 0.013, and 0.012 mm, respectively. The corresponding standard deviations are 0.013, 0.011, 0.01, 0.007, 0.007, 0.007, 0.004, and 0.003, respectively. The average 3D reconstruction accuracy and standard deviation for eight kinds of seeds are 0.017 mm and 0.008 mm, respectively. The visualization of 3D modeling results and the small values of completion errors show that the proposed point cloud completion method is constructive and robust for seed 3D modeling.

### 3.3. Phenotype Estimation Results

Eleven size-related phenotypes and 22 shape-related phenotypes can be automatically extracted based on the reconstructed 3D seed model by automatic trait estimation algorithms. Figure A2 shows seed phenotype results of broad beans, peanuts, pinto beans, soybeans, black beans, red beans, and mung beans. The range, mean values, and the corresponding standard deviation are in detail. It can be noted that different types of seeds have different phenotype values. The shape-related phenotypes have a smaller deviation compared with the size-related phenotypes. The size-related phenotype values of broad beans, peanuts, pinto beans, soybeans, black beans, red beans, and mung beans decrease. The shape-related phenotype values vary due to different phenotype types. 

Figure A3 presents the comparison between the automatically and manually measured seed phenotypes using simple linear regression. It illustrates that size-related phenotypes have higher correlations (*R*^2^ above 0.9981) than shape-related phenotypes (*R*^2^ above 0.8421). The measurement accuracy is to the sub-millimeter.

### 3.4. Shape Description and Quantification Results 

Size- and shape-related statistical models based on the PCA method can be obtained using the proposed method. F_1m_–F_8m_ are seed statistical shape models using 11 size-related phenotypes for broad beans, peanuts, pinto beans, soybeans, black beans, red beans, and mung beans. F_1s_–F_8s_ are seed statistical shape models using 22 shape-related phenotypes.
$$\begin{array}{l} F_{1m}=0.1045X_{01}+0.096X_{02}+0.0893X_{03}+0.1033X_{04}+0.0846X_{05}+0.0683X_{06}+0.1053X_{07}+\\ 0.0656X_{08}+0.0777X_{09}+0.1022X_{010}+0.1033X_{011} \end{array}$$
$$\begin{array}{l} F_{2m}=0.1118X_{01}+0.1039X_{02}+0.1013X_{03}+0.1189X_{04}+0.0819X_{05}+0.1208X_{06}+0.0524X_{07}+\\ 0.0513X_{08}+0.1227X_{09}+0.1087X_{010}+0.0263X_{011} \end{array}$$
$$\begin{array}{l} F_{3m}=0.1042X_{01}+0.1128X_{02}+0.0906X_{03}+0.1119X_{04}+0.084X_{05}+0.0797X_{06}+0.077X_{07}+\\ 0.05X_{08}+0.1005X_{09}+0.0873X_{010}+0.102X_{011} \end{array}$$
$$\begin{array}{l} F_{4m}=0.106X_{01}+0.1051X_{02}+0.1119X_{03}+0.1174X_{04}+0.0745X_{05}+0.0743X_{06}+0.0585X_{07}+\\ 0.1159X_{08}+0.0442X_{09}+0.0925X_{010}+0.0997X_{011} \end{array}$$
$$\begin{array}{l} F_{5m}=0.1055X_{01}+0.1051X_{02}+0.1072X_{03}+0.0837X_{04}+0.1102X_{05}+0.0802X_{06}+0.074X_{07}+\\ 0.0765X_{08}+0.0943X_{09}+0.0953X_{010}+0.0679X_{011} \end{array}$$
$$\begin{array}{l} F_{6m}=0.1083X_{01}+0.1148X_{02}+0.1043X_{03}+0.122X_{04}+0.0864X_{05}+0.102X_{06}+0.0569X_{07}+\\ 0.0386X_{08}+0.0561X_{09}+0.0969X_{010}+0.1137X_{011} \end{array}$$
$$\begin{array}{l} F_{7m}=0.1048X_{01}+0.1067X_{02}+0.107X_{03}+0.1047X_{04}+0.0922X_{05}+0.0819X_{06}+0.0751X_{07}+\\ 0.1121X_{08}+0.0625X_{09}+0.0647X_{010}+0.0884X_{011} \end{array}$$
$$\begin{array}{l} F_{8m}=0.1X_{01}+0.0999X_{02}+0.0993X_{03}+0.0967X_{04}+0.0941X_{05}+0.0902X_{06}+0.0901X_{07}+\\ 0.0886X_{08}+0.0879X_{09}+0.0806X_{010}+0.0726X_{011} \end{array}$$
$$\begin{array}{l} F_{1s}=0.0654X_{1}+0.0491X_{2}+0.0263X_{3}+0.0263X_{4}+0.0683X_{5}+0.0478X_{6}+0.0845X_{7}+\\ 0.0683X_{8}+0.0791X_{9}+0.0449X_{10}+0.0686X_{11}+0.0557X_{12}+0.0109X_{13}+0.0001X_{14}+\\ 0.0348X_{15}+0.0024X_{16}+0.0127X_{17}+0.0146X_{18}+0.0792X_{19}+0.0803X_{20}+0.0406X_{21}+\\ 0.0401X_{22} \end{array}$$
$$\begin{array}{l} F_{2s}=0.0526X_{1}+0.0422X_{2}+0.0473X_{3}+0.0411X_{4}+0.0441X_{5}+0.0488X_{6}+0.0498X_{7}+\\ 0.0397X_{8}+0.0402X_{9}+0.0104X_{10}+0.0516X_{11}+0.0746X_{12}+0.0577X_{13}+0.0717X_{14}+\\ 0.0546X_{15}+0.045X_{16}+0.0711X_{17}+0.0044X_{18}+0.008X_{19}+0.0332X_{20}+0.0563X_{21}+\\ 0.0558X_{22} \end{array}$$
$$\begin{array}{l} F_{3s}=0.0669X_{1}+0.0669X_{2}+0.044X_{3}+0.0869X_{4}+0.0684X_{5}+0.0686X_{6}+0.0545X_{7}+\\ 0.0559X_{8}+0.0111X_{9}+0.0105X_{10}+0.0099X_{11}+0.0581X_{12}+0.0376X_{13}+0.0549X_{14}+\\ 0.0412X_{15}+0.0035XX_{16}+0.0046X_{17}+0.0803X_{18}+0.0288X_{19}+0.0202X_{20}+0.036X_{21}+\\ 0.091X_{22} \end{array}$$
$$\begin{array}{l} F_{4s}=0.0649X_{1}+0.0851X_{2}+0.0538X_{3}+0.0514X_{4}+0.0487X_{5}+0.0449X_{6}+0.0241X_{7}+\\ 0.0339X_{8}+0.0446X_{9}+0.0476X_{10}+0.0472X_{11}+0.0235X_{12}+0.0253X_{13}+0.0243X_{14}+\\ 0.0959X_{15}+0.0949X_{16}+0.0009X_{17}+0.0487X_{18}+0.0503X_{19}+0.0643X_{20}+0.008X_{21}+\\ 0.0177X_{22} \end{array}$$
$$\begin{array}{l} F_{5s}=0.1141X_{1}+0.1406X_{2}+0.1418X_{3}+0.0838X_{4}+0.0025X_{5}+0.0162X_{6}+0.0105X_{7}+\\ 0.02X_{8}+0.0038X_{9}+0.0744X_{10}+0.0086X_{11}+0.0137X_{12}+0.0095X_{13}+0.0075X_{14}+\\ 0.0055X_{15}+0.0173X_{16}+0.1176X_{17}+0.1165X_{18}+0.0298X_{19}+0.0327X_{20}+0.0027X_{21}+\\ 0.0309X_{22} \end{array}$$
$$\begin{array}{l} F_{6s}=0.0532X_{1}+0.0583X_{2}+0.0472X_{3}+0.0564X_{4}+0.066X_{5}+0.064X_{6}+0.026X_{7}+\\ 0.026X_{8}+0.0882X_{9}+0.0898X_{10}+0.0907X_{11}+0.0587X_{12}+0.0085X_{13}+0.0145X_{14}+\\ 0.0449X_{15}+0.0117X_{16}+0.0137X_{17}+0.0387X_{18}+0.0413X_{19}+0.036X_{20}+0.0004X_{21}+\\ 0.0658X_{22} \end{array}$$
$$\begin{array}{l} F_{7s}=0.0398X_{1}+0.0382X_{2}+0.1072X_{3}+0.0784X_{4}+0.105X_{5}+0.0269X_{6}+0.0149X_{7}+\\ 0.0136X_{8}+0.0019X_{9}+0.0582X_{10}+0.0479X_{11}+0.0999X_{12}+0.1002X_{13}+0.0331X_{14}+\\ 0.0344X_{15}+0.0058X_{16}+0.0247X_{17}+0.0323X_{18}+0.0403X_{19}+0.0184X_{20}+0.0475X_{21}+\\ 0.0311X_{22} \end{array}$$
$$\begin{array}{l} F_{8s}=0.0689X_{1}+0.0684X_{2}+0.0538X_{3}+0.0535X_{4}+0.0312X_{5}+0.0784X_{6}+0.0343X_{7}+\\ 0.0345X_{8}+0.0717X_{9}+0.0169X_{10}+0.0514X_{11}+0.0244X_{12}+0.0002X_{13}+0.0676X_{14}+\\ 0.0676X_{15}+0.0052X_{16}+0.0012X_{17}+0.0379X_{18}+0.0685X_{19}+0.0702X_{20}+0.0738X_{21}+\\ 0.0203X_{22} \end{array}$$

Figure 9 and Figure 10 show weight comparisons of different phenotypes in size- and shape-related statistical shape models. It can be found that different phenotypes of different types of seeds play different weights in the same statistical shape model. It can be seen that the weights of the same phenotypes of different species are different and the weights of different phenotypes of the same species also vary. Size- and shape-related statistical models are successfully built, which can provide a possible way to achieve seed shape description and quantification. 

## 4. Discussion

A method for high-throughput seed phenotyping using a handheld 3D laser scanner is presented.

The incompleteness of data often affects the point cloud completion results. Generally speaking, the smaller the incompleteness is, the better the point cloud completion results will be. The results of the least-squares ellipse fitting of the contour line discrete points under different degrees of incompleteness of a peanut seed are shown in Figure 11. It can be seen that the variation of the ellipse fitting results based on the original scanned point cloud, the point cloud data with half of the data missing, and the point cloud data with three-fourths of the data missing is small.

The seed point clouds were obtained in batches with 0.01 mm point cloud density using a handheld 3D laser scanner RigelScan Elite. The ability to handle hundreds of samples in our work in batches outperforms related works [8,10] where dozens of samples are involved. It should be noted that the scanning angle in this work is arbitrary, which makes the data scanning more flexible. Data acquisition in this work still needs a manual operation. However, it is theoretically possible to use the robotic arm to conduct automatic data acquisition with several fixed scanning routes. The handheld 3D laser scanner will combine with the robotic arm to conduct automatic data acquisition in our future work. The experiments show that 3396 of the 3400 samples were successfully scanned. All seeds of broad bean, peanut, soybean, black bean, and mung bean are well scanned except for pinto and red bean seeds. The main factor is the seed surface reflection. The surface reflection of pinto and red bean seeds is heavier than that of broad bean, peanut, soybean, black bean, and mung bean. It should be noted that the background of the image could be changed to obtain more adequate scanning data for different species of seeds theoretically because the working principle of the RigelScan Elite scanner is the triangulation method. In this work, the dark grey background has no changes to conduct data acquisition because the scanned accuracy is high already. Experiments show that the segmentation accuracy is 100%. This high segmentation accuracy is because most of the seeds are isolated, and only a few seeds are sticking and there are no heavily overlapping or pasting seeds.

The complete single-seed 3D model can be directly obtained using an ellipse-based point cloud completion method with an average 3D reconstruction error of 0.017 mm. Figure 12 shows the comparisons of 3D models reconstructed by the screened Poisson surface reconstruction method [24], symmetry-based 3D reconstruction method [33], our proposed method based on the incomplete scanning point cloud, and commercial software (Geomagic Studio) based on the artificially obtained complete scanning point cloud. It shows that the 3D models built by the screened Poisson surface reconstruction method tend to be overfitting and have holes. The symmetry-based 3D reconstruction method only suits seeds with symmetrical shapes, such as soybeans, black beans, red beans, and mung beans, and it fails when seeds have irregular shapes like broad beans, peanuts, and pinto beans. The 3D models built by our proposed method are close to those built by Geomagic Studio. The experiments show that the 3D models built by our method are well fitted and reconstructed. The obtained 3D models are smooth and present the real seed shape well. It can be verified that the proposed 3D modeling method is effective and robust and highlights the related works.

## 5. Conclusions

The objective of the proposed method is to accomplish seed automatic phenotyping for the ideal case, without heavily overlapping and pasting seeds. The goal has been achieved by batch data acquisition using a handheld 3D laser scanner with 0.01 mm point cloud density (99.88% scanning accuracy) and an automatic pipeline of data processing, including single-seed extraction (100% segmentation accuracy), poses normalization, point cloud completion using an ellipse-based point cloud completion method (0.017 mm point cloud completion error), trait estimation (*R*^2^ above 0.9981 for size-related phenotypes and *R*^2^ above 0.8421 for shape-related phenotypes), and statistical models establish based on PCA method (one using size-related phenotypes and the other using shape-related phenotypes). Experiments on eight kinds of seeds with different shapes show that the ability of data acquisition of hundreds of samples, the well-fitted single-seed 3D models, the number and types of the extracted phenotypes, and the unified statistical models for seed shape description and quantification with our proposed method outperform than related works. The ability to handle batch data acquisition and processing, automatic phenotyping, and unified statistical models for seed shape description and quantification, have shown that the proposed method has the potential application for precision agriculture, such as high-throughput seed phenotyping, seed species recognition and classification, and yield trait scorer, etc.

This study can be improved by combining the handheld 3D laser scanner with a robotic arm to conduct automatic data acquisition. It should be noted that there are no heavily overlapping and pasting seeds in our experiment. Further research will explore a more effective segmentation method where seeds are heavily overlapped and pasted. More experiments on seed species recognition and classification using the built statistical models for seed shape description and quantification will be conducted to verify the potential application of the proposed method in the next work.

## Figures and Tables

**Figure 1 sensors-24-06117-f001:**
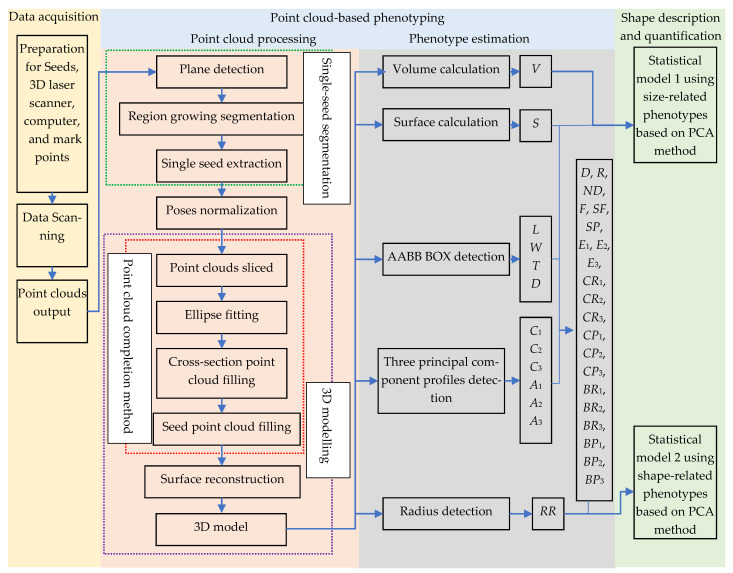
Flowchart for automatic measurement of seed geometric parameters based on handheld scanners.

**Figure 2 sensors-24-06117-f002:**
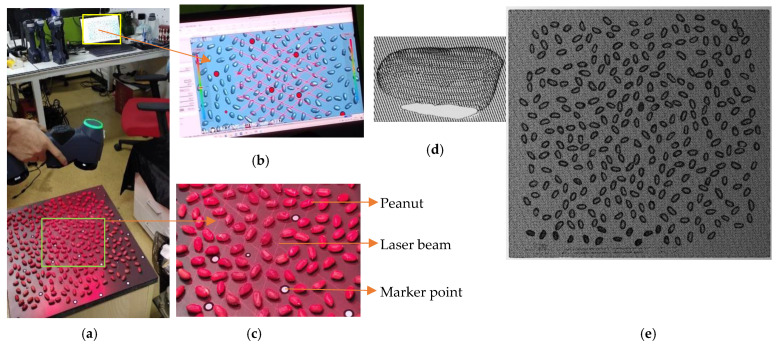
Data acquisition: (**a**) process of data scanning; (**b**) details of obtained point clouds monitoring in real time with rendering visualization; (**c**) details of peanut scanning (red laser crosses are laser beams, and the white points are marker points); (**d**) one sample of the obtained original peanut point cloud, and (**e**) the obtained point clouds of peanuts (filtering 50% for effective visualization).

**Figure 3 sensors-24-06117-f003:**
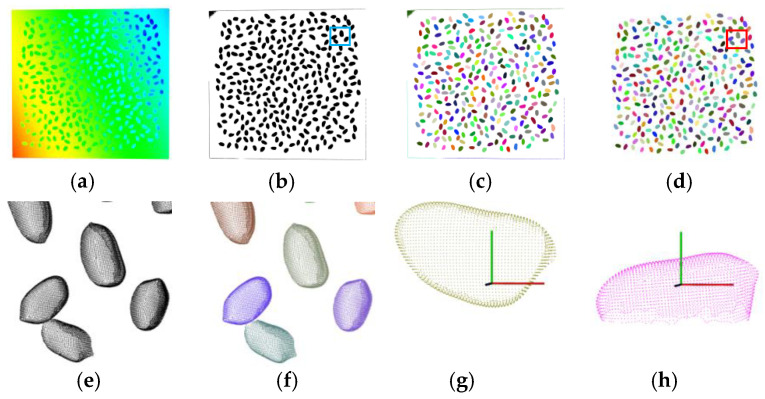
The point cloud processing: (**a**) the scanned point cloud of peanuts; (**b**) the preserved point clouds after RANSAC plane detection; (**c**) the clusters of point clouds after region growing segmentation; (**d**) the single-seed segmentation result; (**e**) details of the scanned point clouds in blue box area in (**b**); (**f**) details of the single-seed segmentation result in red box area in (**d**), and (**g**,**h**) the single peanut seed point cloud in the world coordinate system before and after poses normalization.

**Figure 4 sensors-24-06117-f004:**
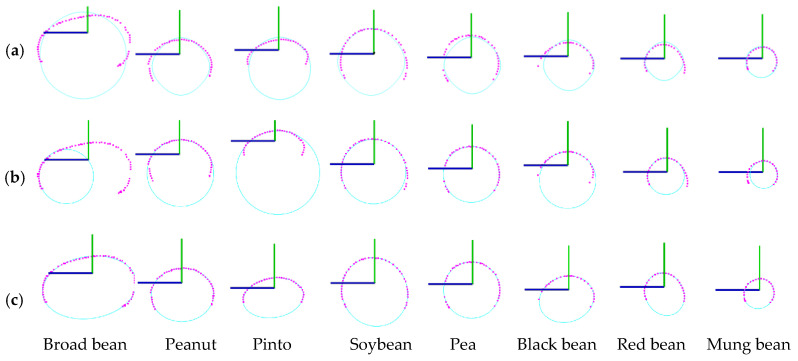
Seed longitudinal profile (*YOZ*) contour fitted by: (**a**) B-spline curve; (**b**) circle, and (**c**) least-squares ellipse.

**Figure 5 sensors-24-06117-f005:**
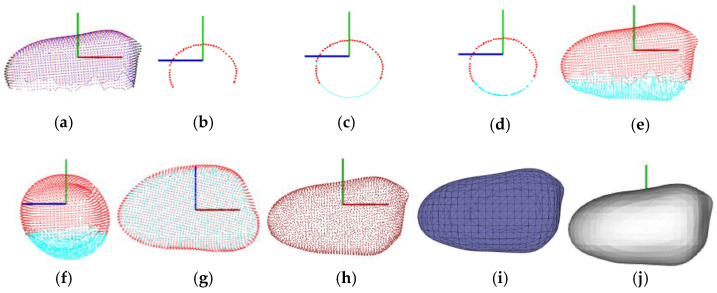
The point cloud processing: (**a**) a series of sliced point clouds; (**b**) one example of the sliced profile point cloud; (**c**) the fitted ellipse (in blue) based on the sliced point cloud; (**d**) the filled complete profile point cloud; (**e**–**g**) one example of the reconstructed peanut point cloud in three view angles, where the red point cloud is the incomplete scanned point cloud and the blue one is the completed point cloud using our proposed ellipse fitting-based point cloud completion method; (**h**) the filtered single peanut seed point cloud, and (**i**,**j**) the triangle mesh and surface visualization of the peanut’s 3D model built by the Poisson surface reconstruction method.

**Figure 6 sensors-24-06117-f006:**
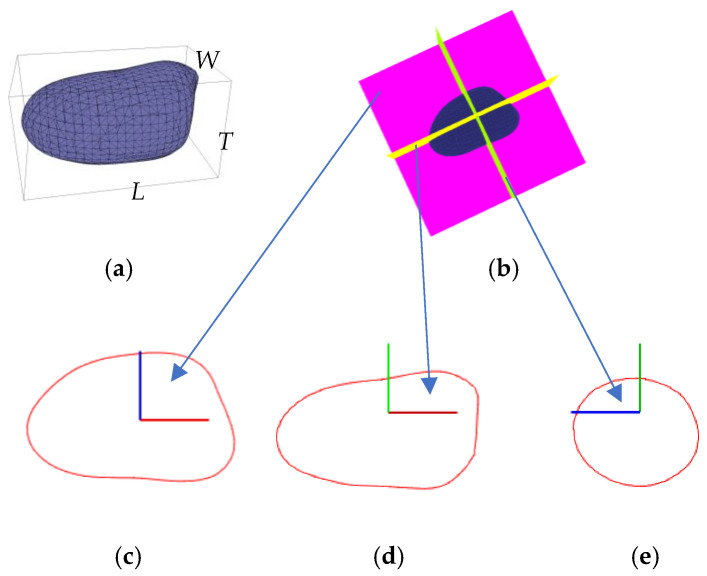
Visualization of size-related phenotypes of a peanut: (**a**) triangulated Poisson mesh, an AABB box of the single peanut 3D model; (**b**) a peanut dividing by three perpendicular principal component profiles (the first one in margarine, second one in yellow, and third one in green), and (**c**–**e**) the first, second, and third profiles.

**Figure 7 sensors-24-06117-f007:**
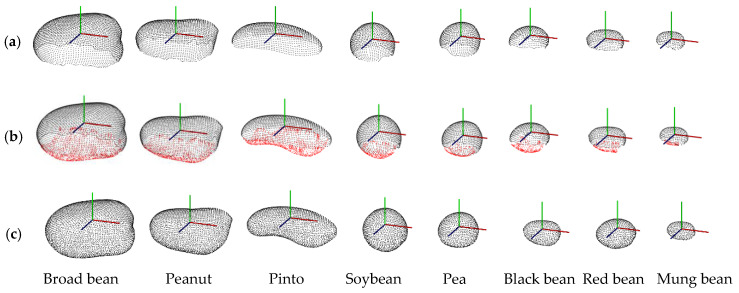
3D point cloud completion results: (**a**) original scanned point clouds; (**b**) completed point clouds, and (**c**) ground truth point clouds.

**Figure 8 sensors-24-06117-f008:**
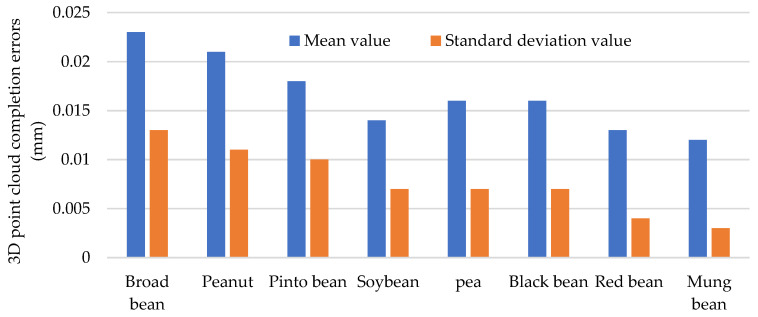
3D point cloud completion errors.

**Figure 9 sensors-24-06117-f009:**
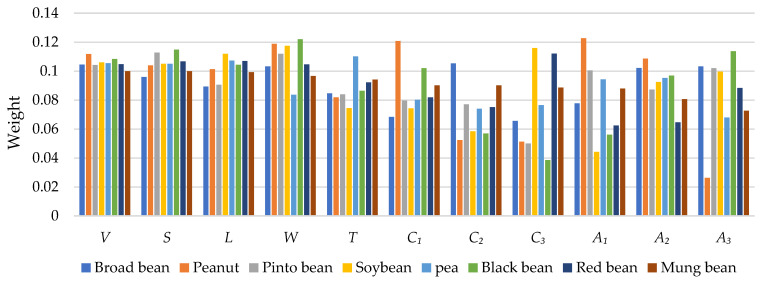
Weight comparisons among 8 types of seeds in the size-related statistical model.

**Figure 10 sensors-24-06117-f010:**
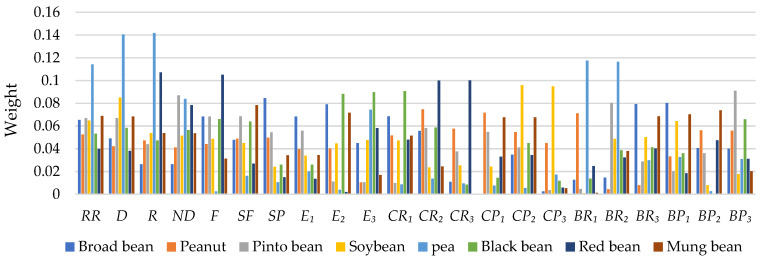
Weight comparisons among 8 types of seeds in the shape-related statistical model.

**Figure 11 sensors-24-06117-f011:**
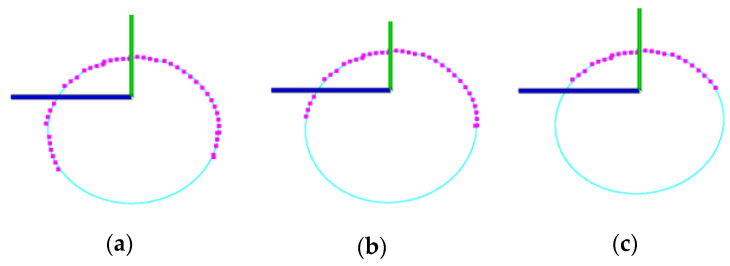
Ellipse fitting based on (**a**) the original scanned point cloud; (**b**) the point cloud with half the data missing; (**c**) the point cloud with three-fourths of the data missing.

**Figure 12 sensors-24-06117-f012:**
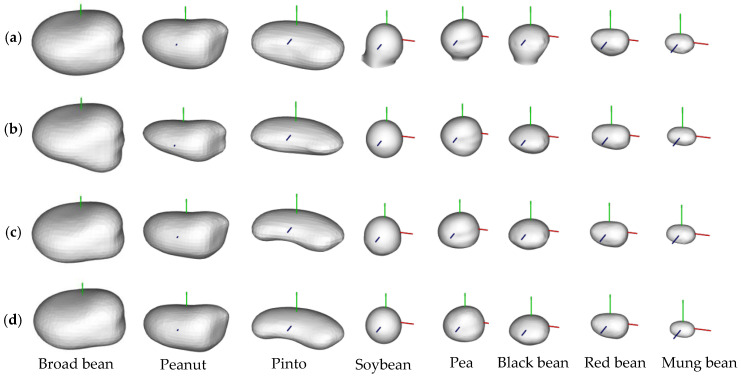
Comparisons of 3D models reconstructed by: (**a**) screened Poisson surface reconstruction; (**b**) symmetry-based 3D reconstruction method; (**c**) our proposed method based on the incomplete scanning point cloud, and (**d**) commercial software (Geomagic Studio) based on the artificially obtained complete scanning point cloud.

**Table 1 sensors-24-06117-t001:** Phenotypes. NO.: the order of Phenotype. Sym.: symbols of Phenotype. Var.: variables.

	NO.	Traits	Sym.	Var.
Size-related phenotypes	1	Volume	*V*	*X* _01_
	2	Surface area	*S*	*X* _02_
	3	Length	*L*	*X* _03_
	4	Width	*W*	*X* _04_
	5	Thickness	*T*	*X* _05_
	6	Horizontal cross-section perimeter	*C* _1_	*X* _06_
	7	Transverse cross-section perimeter	*C* _2_	*X* _07_
	8	Longitudinal cross-section perimeter	*C* _3_	*X* _08_
	9	Horizontal cross-section area	*A* _1_	*X* _09_
	10	Transverse cross-section area	*A* _2_	*X* _010_
	11	Longitudinal cross-section area	*A* _3_	*X* _011_
Shape-related phenotypes	12	Radius ratio	*RR*	*X* _1_
	13	Geometric mean	*D* = (*LWT*)^1/3^	*X* _2_
	14	Roundness	*R* = *L*/(*WT*)^1/2^	*X* _3_
	15	Needle degree	*ND* = *L*/*W*	*X* _4_
	16	Flatness	*F* = *T*/*W*	*X* _5_
	17	Shape factor	*SF* = *TL*/*W^2^*	*X* _6_
	18	Sphericity	*SP* = (*WT*/*L^2^*)*^1^*^/*3*^	*X* _7_
	19	Elongation 1	*E*_1_ = abs((*L* − *W*)/*L*)	*X* _8_
	20	Elongation 2	*E*_2_ = abs((*L* − *T*)/*L*)	*X* _9_
	21	Elongation 3	*E*_3_ = abs((*T* − *W*)/*W*)	*X* _10_
	22	Circularity 1	*CR*_1_ = *C*_1_^2^/4π*A*_1_	*X* _11_
	23	Circularity 2	*CR*_2_ = *C*_2_^2^/4π*A*_2_	*X* _12_
	24	Circularity 2	*CR*_3_ = *C*_3_^2^/4π*A*_3_	*X* _13_
	25	Compactness 1	*CP*_1_ = 16*A*_1_/*C*_1_^2^	*X* _14_
	26	Compactness 2	*CP*_2_ = 16*A*_2_/*C*_2_^2^	*X* _15_
	27	Compactness 3	*CP*_3_ = 16*A*_3_/*C*_3_^2^	*X* _16_
	28	Bounding rectangle 1	*BR*_1_ = *A*_1_/*LW*	*X* _17_
	29	Bounding rectangle 2	*BR*_2_ = *A*_2_/*LT*	*X* _18_
	30	Bounding rectangle 3	*BR*_3_ = *A*_3_/*WT*	*X* _19_
	31	Bounding rectangle to perimeter 1	*BP*_1_ = *C*_1_/2(*L* + *W*)	*X* _20_
	32	Bounding rectangle to perimeter 2	*BP*_2_ = *C*_2_/2(*L* + *T*)	*X* _21_
	33	Bounding rectangle to perimeter 3	*BP*_3_ = *C*_3_/2(*T* + *W*)	*X* _22_

**Table 2 sensors-24-06117-t002:** Scanning and segmentation accuracies.

	*N* _1_	*N* _2_	*N* _3_	Scanning Accuracy (*R*__*scan*_)	Segmentation Accuracy (*R*__*seg*1_)
Broad bean	300	300	300	100.00%	100.00%
Peanut	300	300	300	100.00%	100.00%
Pinto bean	300	299	299	99.67%	100.00%
Soybean	500	500	500	100.00%	100.00%
pea	500	500	500	100.00%	100.00%
Black bean	500	500	500	100.00%	100.00%
Red bean	500	497	497	99.40%	100.00%
Green bean	500	500	500	100.00%	100.00%

## Data Availability

Data and code from this research will be available upon request to the authors.

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
