# Peer review of "Automatic Measurement of Seed Geometric Parameters Using a Handheld Scanner"

_sensors, 2024, doi:10.3390/s24186117_

Round 1

Reviewer 1 Report

Comments and Suggestions for Authors

The paper seems to fit well with the Special Issue Novel Sensors for Precision Agriculture Application in Sensors. The paper focuses on proposing a novel method for automatic measurement of three-dimensional seed phenotypes. Thus, the authors have added a new knowledge to the previously known subject area described in other published material. The level of originality of the paper is good and the paper could be classified as quite interesting.

In the introduction, the authors have provided the main purpose of the study and the background to the topic, which includes many relevant references. The methodology section is well written and thorough; however, the part of the text containing the equations needs to be completed. The authors have also provided experimental results, which have been discussed. The conclusions are consistent with the evidence and arguments presented earlier.

I also have some detailed comments, which are listed below:

1.        Please avoid introducing abbreviations without giving full names, e.g., R2 on page 1, line 23, GPU on page 2, line 82.

2.        Page 1, line 24. Please write “R2” in superscript instead of “R2”. Page 11, line 338. Please write “zij” in superscript instead of “zij”.

3.        Page 11. All matrices and vectors should be marked in bold, not in italics.

4.        Page 11, lines 334-335. Please clarify this part.

5.        Page 11, line 338. Please check the correctness of the equation of zij. In my opinion, first sj should be changed to yij.

6.        Page 11, eq. 11. Please define λ (only λ1, λ2, λ3 were defined before) and Ip.

7.        Page 11, line 343. How do you calculate m?

8.        Page 11, eq. (13). Please define bj0.

9.        Page 12, line 348. Should there be n here? n was defined earlier as the number of rows in the matrix X.

10.    Page 12, eq. (17). Please define pij.

11.    Page 12, eq. (19). n had a different meaning before (see remark 8). Please do not repeat the designations with different meanings.

12.    Figures 10 and 11 are difficult to read. Subfigures and font should be larger. For example, 1) use a common legend for the whole figure, not separate but the same legend for each subfigure, or 2) provide four figures instead of two figures.

13.    Page 20, line 484. It should be “Figures 12 and 13”.

Author Response

Thank you immensely for dedicating your time to review this manuscript. Please find the detailed responses outlined below. The revised manuscript is attached for your convenience.

Point 1:  Please avoid introducing abbreviations without giving full names, e.g., R2 on page 1, line 23, GPU on page 2, line 82.

Response 1: Many thanks for this valuable comment. The problem has been solved.

We have corrected this error in lines 23 and 84 of the revised manuscript.

Point 2: Page 1, line 24. Please write “R2” in superscript instead of “R2”. Page 11, line 338. Please write "zij" in superscript instead of "zij".

Response 2: Thanks very much for your constructive comment. This problem has been solved.

We have addressed and rectified this error in lines 24 and 345 of the revised manuscript. 

Point 3: Page 11. All matrices and vectors should be marked in bold, not in italics.

Response 3: Thank you very much for your valuable opinions. This mistake has been solved.

The matrices and vectors have been marked in bold of the revised manuscript.

(Page 11 and 12, lines 338,340,343,346,347,348,355,358,359,and 360)

Point 4: Page 11, lines 334-335. Please clarify this parts.

Response 4: Thank you very much for your valuable suggestion.  This error has been solved.

We have clarified this parts in lines 335 and 336 of the revised manuscript.

Point 5: Page 11, line 338. Please check the correctness of the equation of zij. In my opinion, first sj should be changed to yij. 343

Response 5: Thank you very much for your valuable opinions.  This error has been solved.

In the  the equation of zij in line 344 of the revised manuscript, the first sj have be changed to yi.

Point 6: Page 11, eq. 11. Please define λ (only λ1, λ2, λ3 were defined before) and Ip.

Response 6: Thank you very much for your valuable suggestion.  This error has been solved.

We have defined λ and Ip  in lines 348 and 349 of the revised manuscript.

Point 7: Page 11, line 343. How do you calculate m?

Response 7: Thank you very much for your valuable question.  The value of is accumulated contribution rate. When the value  of  is greater than 80%, we can see that the number of new features is m. Thus m can be calculated.

Point 8: Page 11, eq. (13). Please define bj0.

Response 8: Thank you very much for your valuable comment.  The problem has been solved.

We have defined it in line 354 of the revised manuscript.

Point 9:  Page 12, line 348. Should there be n here? n was defined earlier as the number of rows in the matrix X.

Response 9: Thank you very much for your valuable question.  This error has been solved.

wi instead of n should be here. We have corrected this problem in line 357 of the revised manuscript.

Point 10: Page 12, eq. (17). Please define pij.

Response 10: Thank you very much for your valuable suggestion.  This problem has been solved.

We have defined it in line 367 of the revised manuscript.

Point 11: Page 12, eq. (19). n had a different meaning before (see remark 8). Please do not repeat the designations with different meanings.

Response 11: Thank you very much for your valuable suggestion.  This error has been solved.

We have used  np instead of n  in lines 384 and 385 of the revised manuscript.

Point 12: Figures 10 and 11 are difficult to read. Subfigures and font should be larger. For example, 1) use a common legend for the whole figure, not separate but the same legend for each subfigure, or 2) provide four figures instead of two figures.

Response 12: Thank you very much for your valuable comment.  This error has been solved.

To enhance the readability of Figures 10 and 11 in the original manuscript, we have incorporated your insightful feedback and the perspectives of another reviewer. Consequently, we have substituted these figures with higher-resolution images, which are now available in Appendix A2 and A3. Additionally, we have updated the names of the corresponding figures in the revised manuscript, as indicated in the highlighted sections in the rest of the revised manuscript.

(Pages 21 and 23, lines 598-602).

Point 13: Page 20, line 484. It should be “Figures 12 and 13”.

Response 13: Thank you very much for your valuable question.  This error has been revised.

We have corrected this error in line 484  of the revised manuscript.

Reviewer 2 Report

Comments and Suggestions for Authors

In this work, the scientific team proposed a method for the automatic phenotyping of seeds of asymmetric shapes. During the research, 8 types of seeds were tested.

In the introduction, the authors outlined the main goals and objectives of the study, provided a rationale, and the necessity for developing this phenotyping technology. The main advantage of the proposed method is the reduction of time for seed phenotyping, due to the automation in data collection and processing.

The 'Materials and Methods' section presents detailed materials on the principles and algorithms for collecting and processing the geometric parameters of plant seeds, and describes in detail the proposed phenotyping method 'Point Cloud-based Phenotyping'.

The 'Results' and 'Discussion' sections present the results of the test measurements, as well as an assessment of the effectiveness of the proposed method. The main disadvantages and difficulties encountered during the test trials are indicated.

While reading the article, several questions arose:

1.       It is not entirely clear what the fundamental difference is from the work previously done (doi:10.3390/rs14020431), as the main algorithms and methods have been transferred. Most of the formulas coincide; it might have been sufficient to provide a reference and not describe them again, but only to indicate the changes introduced to the method.

2.       Perhaps it would be advisable to move some of the graphical materials (figures 7, 10, 11) to the appendix, leaving only the most necessary, as well as the polynomial formulas.

3.       In the discussion, it was mentioned that there were problems with scanning pinto and red bean seeds, but it is not entirely clear how this problem was resolved during the test measurements.

4.       During the measurement process, there are moments when non-standard seeds or seeds of other crops are encountered. How will this issue be addressed in such cases?

5.       The conclusions draw attention to the fact that: "It should be noted that there are no heavily overlapping and pasting seeds in our experiment. Further research will explore a more effective segmentation method where seeds are heavily overlapped and pasting." A similar conclusion was made in the previous work "In addition, there are no overlapping and pasting seeds in our experiment. Further research will explore an effective segmentation method where seeds are overlapping and pasting." It might be worth noting that such separation cannot be achieved by software means alone and that preliminary mechanical processing of the test material is required.

Author Response

Thank you immensely for dedicating your time to review this manuscript. Please find the detailed responses outlined below. The revised manuscript is attached for your convenience.

Point 1: It is not entirely clear what the fundamental difference is from the work previously done (doi:10.3390/rs14020431), as the main algorithms and methods have been transferred. Most of the formulas coincide; it might have been sufficient to provide a reference and not describe them again, but only to indicate the changes introduced to the method.

Response 1: Many thanks for your constructive  opinions. The article with the DOI 10.3390/rs14020431 serves as our foundation, and this current paper represents a significant leap forward, introducing three notable advancements.

(1) In our previous article, we relied solely on symmetry to accomplish the point cloud reconstruction of seeds. Nevertheless, in this article, we adopt a novel point cloud completion method based on ellipse fitting and smooth interpolation to reconstruct the point clouds of seeds. While the previous approach was limited to seeds with symmetrical geometric shapes, the current method is capable of handling point cloud data reconstruction for seeds with irregular geometric shapes. As a result the point cloud completion algorithm presented herein is superior, boasting greater universality and robustness.

(2) The geometric parameter measurement precision for seeds achieved in this article is higher than that in our previous article.

(3) In addition to measuring geometric parameters, this paper utilizes those measurements to further explore and quantify the shape description of seeds, whereas the previous article only discussed the measured geometric parameters.  is our   previous work. This paper makes three significant advancements compare to it.

Point 2: Perhaps it would be advisable to move some of the graphical materials (figures 7, 10, 11) to the appendix, leaving only the most necessary, as well as the polynomial formulas.

Response 2: Thank you very much for your valuable comment.  This problem has been solved.

Figures 7, 10, and 11 from the initial draft have been moved to Appendices A1, A2, and A3 in the  revised manuscript. Additionally, we have updated the names of the corresponding figures in the revised manuscript, as indicated in the highlighted sections in the rest of the revised manuscript.

(Pages 20-23, lines 592-602)

Point 3: In the discussion, it was mentioned that there were problems with scanning pinto and red bean seeds, but it is not entirely clear how this problem was resolved during the test measurements.

Response 3: Thank you very much for this comment. Due to the unique physical properties of the reflective surfaces of pinto and red bean seeds, their scanning accuracy tends to be lower compared to other seed types. However, as the primary focus of this article does not entail enhancing seed scanning accuracy, we have disregarded this matter and instead concentrated our efforts on developing a method for filling in incomplete scanning data of seeds.

Point 4: During the measurement process, there are moments when non-standard seeds or seeds of other crops are encountered. How will this issue be addressed in such cases?

Response 4: Thank you very much for your question. We are sorry for not grasping your query in its entirety. If you're inquiring about the measurement of seeds that exhibit non-standard physical geometric shapes, such as length, width, and height, we utilize a vernier caliper to precisely determine the dimensions of the geometric center of the seeds. These measurements serve as our ground truths. Additionally, for other parameters ground truths, we rely on specialized software (Geomagic Studio) to individually measure them based on the complete 3D model of each seed. To obtain this model, we affix a single seed to a desktop using a needle, scan it comprehensively in all directions, and then preprocess the resulting 3D model using Geomagic Studio. For more detailed information, please refer to our previous article  (doi:10.3390/rs14020431).

If you're asking about how our algorithm handles seeds with non-standard geometric shapes, we have conducted experiments using seeds like broad beans, pinto beans, and peanuts. After rigorous testing, we can confirm that our algorithm is well-suited for measuring seeds with irregular or non-standard geometric shapes.

Point 5: The conclusions draw attention to the fact that: "It should be noted that there are no heavily overlapping and pasting seeds in our experiment. Further research will explore a more effective segmentation method where seeds are heavily overlapped and pasting." A similar conclusion was made in the previous work "In addition, there are no overlapping and pasting seeds in our experiment. Further research will explore an effective segmentation method where seeds are overlapping and pasting." It might be worth noting that such separation cannot be achieved by software means alone and that preliminary mechanical processing of the test material is required.

Response 5: We appreciate your insightful comment immensely. We have conducted extensive research by consulting numerous sources and methods, seeking to devise a strategy for segregating individual seeds that overlap and adhere to each other. We envision leveraging the power of deep learning techniques to achieve a more intricate and precise point cloud completion. While we are currently engaged in this research and have yielded promising experimental outcomes, we recognize that further exploration is crucial. However, we also ponder whether this intricate process is absolutely necessary. As you have aptly pointed out, we may be able to significantly reduce the occurrence of severe seed overlap through physical separation methods. This approach could potentially simplify our workflow while still achieving our desired objectives. Thank you again for your valuable feedback.

Reviewer 3 Report

Comments and Suggestions for Authors

The article describes a new method for automatically measuring three-dimensional seed phenotypes, which is an urgent task. Eight types of seeds were tested with 3,400 samples. The authors used a handheld 3D laser scanner and an automated data processing pipeline.

The introduction provides sufficient background, includes ample references to the literature, and materials and methods are described in detail.

Questions & Mistakes:

1. The manuscript looks overloaded with graphs and diagrams. Figures 10, 11 can be designed as additional materials for the article.

2. Previously, the authors have already proposed a method for phenotyping legume seeds of symmetrical shape: soybeans, peas, black beans, red beans and mung beans. Why did the authors choose symmetrical seeds for research in this work, since the main difference of the new method is the possibility of 3D reconstruction of asymmetrical seeds of complex shape?

3. The main limitation of the proposed method is that the experiment did not involve heavily overlapping and pasting seeds. The authors indicated that this will be explored later. In the previous 2022 paper, the authors also indicated that experiments with heavily overlapping and pasting seeds would be carried out later, but such experiments were not carried out in the new study. Also, it is not entirely clear how the accuracy of measurements can be affected by debris and other seed contamination. Thus, the obtained scanning accuracy values ​​are valid only for the ideal case, which is rarely encountered in real work on seed phenotyping. In the goals of the work, it should be added that the technique is developed for the ideal case, without heavily overlapping and pasting seeds.

Author Response

Thank you immensely for dedicating your time to review this manuscript. Please find the detailed responses outlined below. The revised manuscript is attached for your convenience.

Point 1: The manuscript looks overloaded with graphs and diagrams. Figures 10, 11 can be designed as additional materials for the article.

Response 1: Thank you very much for the valuable suggestions and comments. This problem has been solved.

Figures 7, 10, and 11 from the initial draft have been moved to Appendices A1, A2, and A3 in the  revised manuscript. Additionally, we have updated the names of the corresponding figures in the revised manuscript, as indicated in the highlighted sections in the rest of the revised manuscript.

(Pages 20-23, lines 592-602)

Point 2: Previously, the authors have already proposed a method for phenotyping legume seeds of symmetrical shape: soybeans, peas, black beans, red beans and mung beans. Why did the authors choose symmetrical seeds for research in this work, since the main difference of the new method is the possibility of 3D reconstruction of asymmetrical seeds of complex shape?

Response 2: Many thanks for this question. By selecting symmetric seeds (soybeans, peas, black beans, red beans, and mung beans) for our research, we aim to establish a robust benchmark or reference model that can facilitate a deeper understanding and application of novel methods to asymmetric seeds (broad beans, peanuts, and pinto beans), which often exhibit more complex shapes. This study of symmetric seeds serves as a theoretical foundation and technological backbone for the 3D reconstruction of asymmetric seeds, ultimately contributing to the advancement and refinement of plant phenotype research. The experiment shows that the method proposed in this article is applicable to geometrically symmetric and asymmetric seeds.

Point 3: The main limitation of the proposed method is that the experiment did not involve heavily overlapping and pasting seeds. The authors indicated that this will be explored later. In the previous 2022 paper, the authors also indicated that experiments with heavily overlapping and pasting seeds would be carried out later, but such experiments were not carried out in the new study. Also, it is not entirely clear how the accuracy of measurements can be affected by debris and other seed contamination. Thus, the obtained scanning accuracy values are valid only for the ideal case, which is rarely encountered in real work on seed phenotyping. In the goals of the work, it should be added that the technique is developed for the ideal case, without heavily overlapping and pasting seeds.

Response 3: Many thanks for this suggestion. This problem has been solved. We have revised the closing statement of your introduction.

The revised parts are on page 19 in line 550 of the revised manuscript.

Round 2

Reviewer 2 Report

Comments and Suggestions for Authors

Good afternoon.

The explanations received answered most of the questions that arose.

Please consider these minor comments:

1. Formulas 5 and 6 contain non-correct symbols "".

2. Perhaps it is worth transferring the polynomial equations to the application.

Author Response

Thank you immensely for dedicating your time to review this manuscript. Please find the detailed responses outlined below. The revised manuscript is attached for your convenience.

Point 1: Formulas 5 and 6 contain non-correct symbols ?

Response 1: Many thanks for your constructive  comment. This problem has been solved.

Formulas 5 and 6 have been changed with correct symbols in the revised manuscript.

Point 2: Perhaps it is worth transferring the polynomial equations to the application.

Response 2: Thank you very much for your valuable suggestion.  One of the innovations of our paper is to propose shape description and quantification based on statistical models, so we have retained these polynomials. In the next step of our research, we will consider utilizing the description and quantification of seed geometric shapes, namely these polynomials, for classification applications.

Reviewer 3 Report

Comments and Suggestions for Authors

The authors supplemented and corrected the manuscript. I have no more questions.

Author Response

Comment: The authors supplemented and corrected the manuscript. I have no more questions.

Response: 

Once again, I would like to express my sincere gratitude for your valuable suggestions and opinions in the first round. Your insights have greatly inspired me.